# Potential sources of contamination on textiles and hard surfaces identified as high-touch sites near the patient environment

Erik Nygren[1], Lucia Gonzales Strömberg[1]*, Jenny Logenius[2], Ulrika Husmark[2], Charlotta Löfström[1], Birgitta Bergström[1]

**1** RISE Research Institutes of Sweden, Agriculture and Food, Borås, Sweden, **2** Essity Hygiene and Health AB, Gothenburg, Sweden

* lucia.gonzales.stromberg@ri.se

**Data Availability Statement:** All 26 genome sequences files are available from the NCBI database with project ID PRJNA924984 with the

## Abstract

The hospital environment represents an important mediator for the transmission of health-care-associated infections through direct and indirect hand contact with hard surfaces and textiles. In this study, bacteria on high-touch sites, including textiles and hard surfaces in two care wards in Sweden, were identified using microbiological culture methods and 16S rDNA sequencing. During a cross-sectional study, 176 high-touch hard surfaces and textiles were identified and further analysed using microbiological culture for quantification of total aerobic bacteria, *Staphylococcus aureus*, *Clostridium difficile* and Enterobacteriacae. The bacterial population structures were further analysed in 26 samples using 16S rDNA sequencing. The study showed a higher frequency of unique direct hand-textile contacts (36 per hour), compared to hard surfaces (2.2 per hour). Hard surfaces met the recommended standard of $\leq 5$ CFU/cm$^2$ for aerobic bacteria and $\leq 1$ CFU/cm$^2$ for *S. aureus* (53% and 35%, respectively) to a higher extent compared to textiles (19% and 30%, respectively) (P = 0.0488). The number of bacterial genera was higher on textiles than on the hard surfaces. *Staphylococcus* (30.4%) and *Corynebacterium* (10.9%) were the most representative genera for textiles and *Streptococcus* (13.3%) for hard surfaces. The fact that a big percentage of the textiles did not fulfil the criteria for cleanliness, combined with the higher bacterial diversity, compared to hard surfaces, are indicators that textiles were bacterial reservoirs and potential risk vectors for bacterial transmission. However, since most of the bacteria found in the study belonged to the normal flora, it was not possible to draw conclusions of textiles and hard surfaces as sources of healthcare associated infections.

## Introduction

Infectious diseases are a leading cause of morbidity and mortality in society and healthcare systems worldwide [1]. Infection control interventions are important to reduce the spread of infectious diseases. Hand hygiene and disinfection, as well as microbial decontamination of high-touch surfaces, including textiles and hard surfaces, are considered key infection control measures [2–6]. Even though more emphasis has been placed on the disinfection of hard

following accession numbers SAMN32777245
SAMN32777246 SAMN32777247
SAMN32777248 SAMN32777249
SAMN32777250 SAMN32777251
SAMN32777252 SAMN32777253
SAMN32777254 SAMN32777255
SAMN32777256 SAMN32777257
SAMN32777258 SAMN32777259
SAMN32777260 SAMN32777261
SAMN32777262 SAMN32777263
SAMN32777264 SAMN32777265
SAMN32777266 SAMN32777267
SAMN32777288 SAMN32777299
SAMN32777270.

**Funding:** Vinnova, Sweden's innovation agency,
grant no. 2014-00719 to BB, JL, UH. https://www.
vinnova.se/ The funding agency did not play any
role in the study design, data collection, analysis,
or preparation of the manuscript.

**Competing interests:** The authors have declared
that no competing interests exist.

surfaces, the decontamination of soft textiles is considered equally important and therefore needs to be addressed [7].

Healthcare associated infections (HCAI), or nosocomial infections, occur in patients under medical care, as well as in health care workers. HCAI is associated with substantially prolonged hospital stay, long-term disability, increased antibiotic resistance, increased mortality and lead to suffering and economic consequences [8]. Although many HCAI are preventable through good hand hygiene, several studies indicate that environmental cleaning practices and microbial decontamination also play significant roles in transmission of HCAI [9–11].

Microorganisms commonly associated with HCAI, such as methicillin-resistant *Staphylococcus aureus* (MRSA), vancomycin-resistant enterococci (VRE), *Clostridioides difficile (*previously *Clostridium difficile)*, *Acinetobacter* spp. and norovirus, can persist on surfaces and remain infectious for hours to days (and in some cases even for months) [12, 13]. HCAI-associated microorganisms have been found to be transmitted through direct and indirect hand contact with hard and non-porous surfaces, such as technical devices, bed rails, bedside tables, and floors [8, 14, 15]. Further, porous surfaces and textiles, e.g., bed linen and clothing worn in healthcare settings, have also been suggested in the acquisition and transmission of these microorganisms [16, 17]. However, little is known about the extent to which these materials are contaminated and are a contributing factor for an increased risk of HCAI [7].

Studies investigating the role of inanimate objects in the transmission of HCAIs and infection control strategies have primarily focused on medical instruments and high-touch non-porous objects and less on soft surfaces and healthcare textiles [7]. Thus, to minimize transmission of microorganisms, it is important to identify high-touch objects that can be related to the transmission routes [18]. There are standards to monitor the quality of room cleaning and disinfection, to ensure that surfaces have been treated appropriately. The monitoring strategies include simple visual inspections, microbiological testing of surface contamination and technological innovations that measure the adequacy of surface cleaning [19]. Among the microbiological testing, an acceptance level of less than five colony-forming units (CFU) of aerobic microorganisms per $cm^2$ on surfaces [20, 21] and $< 1$ $CFU/cm^2$ for bacteria such as *S. aureus* and *Enterococcus* is used to assess the cleanliness of a surface, following the Swedish standard SS 8760014:2017 [22].

The aim of this study was to identify bacteria present on high-touch sites, including textile fomites and hard surfaces, in one neonatal and one cardiac care ward using microbiological culture methods and 16S rDNA sequencing.

## Methods

### Study design

A cross-sectional study and sample collection was conducted in Stockholm, Sweden, at one cardiac care ward at Capio St Göran Hospital from January to March 2015 and at one neonatal care ward at Stockholm South General Hospital (Södersjukhuset) in May 2015.

The study was performed to reveal the nature and frequency of contacts between hospital textile and patients. A study protocol was designed including the following criteria: 1) Direct textile contact, defined as skin contact with a textile; 2) Indirect textile contact, defined as skin contact with a surface that has been cleaned with a multi-use textile; 3) Person category, registering to which category the observed person belonged, i.e. patient, nurse, etc.; 4). Duration, a binary category including brief contact or extended exposure. The observation was performed by the same person, three times during a two-hour session per day, for nine days at the neonatal care, and for five days at the cardiac ward. The surfaces identified as high-touch surfaces were sampled for further analysis. Factors such as textile or surface types, time span,

behavioural and person categories were included. The samples collected at both hospitals were handled as one set of samples for the analysis.

### Ethical approval

The observational study was approved by the ethics committee of each hospital under the premises that no interventions would be conducted that would affect the physical and psychological safety of the patients or the staff; and that the patients would be informed in advance of the study and would have the right not to participate. Surface and fabric samples collected in this study did not qualify as biological samples according to the Swedish ethical law.

### Sample collection

One hundred and seventy-six environmental samples from textiles and surfaces, previously cleaned according to hospital cleaning guidelines with textile-based cleaning materials, were collected and analysed regarding the number of bacteria and the bacterial community composition. The samples for hard surfaces included swabs from furniture, bathroom surfaces, and the floor in both the patient room and the bathroom, whereas the textiles samples included rinses of bed linen, pillowcases, nursery beds, hospital clothes, faucets, drapes and floor mops. All surfaces were in the patient close environment or were other surfaces that the patient could come in contact with.

In the study, triplicate samples were collected from identified high-touch objects/surfaces during within 20 minutes of observing the contact. For the hard surfaces, samples were taken by swabbing 10 cm$^2$ of the surfaces, using a pre-moistened sampling sponge (HS10NB2G, 3M™ Hydrated-Sponge) containing a neutralizing buffer [23]. The sponge was swiped across a surface vertically with one side, then horizontally with the other side, then diagonally using the sides of the sponge and lastly the tip of the sponge was applied around the edges. Sponges were then placed into sealable stomacher bags (BagLight 400 PolySilk; Interscience) and stored at 4˚C until analysis. To verify the sterility of the sponges, a clean sponge moistened with the neutralizing buffer was included as a negative control. Textile samples (10 x 10 cm) were collected by rinsing clothing and sheets using peptone water according to the protocol [24]. All collected samples were processed within 24 h of sampling, similarly, as described [23]. To keep variability of the sampling technique to a minimum all sampling was performed by the same individual.

### Sample preparation

The collected sponges and textiles were soaked in saline buffer 0.9%, with total volumes of 15 and 50 ml, respectively. Following stomaching for 180 seconds, sample fluid was recovered from the sponge or textile by pressing by hand from the outside of the Stomacher bags. After collection, the sample fluids were centrifuged at 5 000 × g for 15 min, the supernatant was discarded and the pellet was resuspended in 300 µl of peptone water (Peptone Bacteriological Neutralized, Oxoid). A volume of 100 µl was immediately frozen at—80˚C for later DNA extraction and the remaining 200 µl was used for the microbial culture analysis.

### Microbial culture analysis

One hundred microliter of the recovered suspension was serially diluted for quantification of the total aerobic bacteria using TSA agar plates (Tryptone Soya Agar, Oxoid), Enterobacteriaceae as an indicator of faecal bacteria using Violet Red Bile Glucose (VRBG) plates (Bacto VRBGA; Oxoid) and *S. aureus* using Baird Parker Agar (Baird Parker-RPF agar, Biomerieux).

The remaining sample volume of ca 100 µl was inoculated onto *Clostridium difficile* Selective Agar (BBL C. *difficile* Selective Agar, Becton Dickinson) and incubated anaerobically for the detection of *C. difficile*. Because of some differences in the actual sample surface sizes all CFU counts were normalized to 10 $cm^2$ before data analysis. To be able to assess the cleaning efficiency, the acceptance level criteria was set to $\leq 5$ $CFU/cm^2$ for total aerobic microorganisms and $\leq 1$ $CFU/cm^2$ for *S. aureus* and *C. difficile* [22].

**DNA extraction and 16S rDNA amplicon sequencing.** Genomic DNA from the environmental swab samples and textiles were extracted using Qiagen DNeasy Blood & Tissue Kit according to the manufacture's protocol for gram positive bacteria, with an elution volume of 100 µl. A negative control (sample collection buffer) was included during the extraction and processed with the samples. All extracted DNA was quantified using the Qubit double-stranded DNA (dsDNA) high-sensitivity kit (Thermo Fisher Scientific). Genomic libraries were prepared using Illumina Nextera DNA library prep according to the manufacturer's protocol and were sequenced using the Illumina MiSeq system, thereby obtaining paired-end 150 bp reads (Eurofins Genomics, Konstanz, Germany).

## Bioinformatics and statistical analysis

Publicly available sequence data, i.e. 24 faecal samples from healthy human controls collected in two studies (PRJNA259188, and PRJNA354863) [25]; 28 surface samples from a neonatal intensive care unit (PRJNA432186) [26] and 80 skin samples (foot and hand) from healthy humans (PRJNA345497), were downloaded from NCBI in May 2018. These raw sequences were analysed together with our raw sequencing data. All sequences were subjected to adapter trimming, quality trimming and filtering (Q30) via kmer matching using BBDuk [27]. Clustering, classification and quantification of "operational taxonomic units" (OTUs), i.e. 16S rDNA genes, were performed on the Nephele platform [28], using QIIME v1.9.1 [29]. Chimeras were removed and the open reference OTU picking process were applied at 99% against the Greengenes 13_8 99% reference database [30]. Prior to comparisons between the different sample groups, filtering out of non-bacterial 16S rDNA genes was performed as described [31]. MicrobiomeAnalyst was used for the analysis and graphics of the alfa-diversity analysis [32, 33].

The statistical analysis including chi-squared calculation and unpaired two-tailed t-test using a significant value of $p < 0.05$ were performed using GraphPad Prism 8.3.0.

## Results

Results showed that the frequency of unique direct hand-textile contacts at the neonatal ward was approximately 36 per hour (1000 over 28 h) while there were only 4 per hour contacts (200 over 54 h) at the cardiac ward. The frequencies of contacts between the skin and recently cleaned surfaces were less frequent, 2.2 occasions per hour (120 over 54 h) at the cardiac care ward and 0.5 occasions per hour (13 over 28 h) at the neonatal ward.

Direct skin contacts at the neonatal ward were found to be mainly contacts with the textiles in and around the incubators (75% of contacts). The direct contacts at the cardiac care ward were dominated by the textiles around the patient's bed, with the majority of contacts associated with handling the drapes (67%). At the cardiac care ward, nurses were responsible for most of the direct textile contacts, prominently with the drapes. This corresponds to a total of 720 direct contacts with the drapes during the study period; changing of drapes occurs every 90 days.

## Microbial sampling and culturing of bacteria

Microbiological sampling was conducted at multiple textiles and hospital equipment (including furniture), identified as high hand touch surfaces, during the time of the study. Sixty-three

**Table 1. Number of surfaces out of the 176 totally analysed surfaces that that met the criteria for microbiological cleanliness for aerobic bacteria and *S. aureus*, respectively.**

| Surface | Aerobic bacteria | *S. aureus* |
|---|---|---|
|  | $< 5$ CFU/cm$^2$ | $< 1$ CFU/cm$^2$ |
|  | n (%) | n (%) |
| Textiles | 33 (18.7) | 53 (30.1) |
| Hard Surfaces | 93 (52.8) | 62 (35.2) |
| Total | 126 (71.6) | 115 (65.3) |

samples from textiles and 113 samples from hard surfaces were collected. Microbial culture analysis was performed to determine the presence of aerobic bacteria, *S. aureus*, Enterobacteriaceae and *C. difficile*. Hard surfaces met the recommended standard of $\leq 5$ CFU/cm$^2$ for aerobic bacteria (p = 0.0488, using Chi-square test) and $\leq 1$ CFU/cm$^2$ for *S. aureus*, to a greater extent than textiles (Table 1). No *C. difficile* was found in neither the textiles nor the hard surfaces.

## 16S rDNA sequencing analysis

In addition to the culturing of bacteria from the sampled areas, sequencing of the bacterial 16S rDNA genes was conducted. Samples coming from the same high-touch objects/surfaces group were pooled for the analysis and were divided in 26 different groups, including 10 samples from textiles (i.e., all textiles and non-woven materials) and 16 samples from hard surfaces (i.e., tables, alarms and technical devices). The DNA sequences are available at NCBI with ID-PRJNA924984 (S1 Table). For comparison, the bioinformatics and core microbiome analysis also included 132 publicly available data sets from four similar 16S rDNA sequencing studies with faecal samples, surface samples from a neonatal intensive care unit and skin samples.

The quality control and length filtering of the raw 16S rDNA reads obtained from the 26 samples resulted in a total of 114,875,672 (mean: 4,994,594; range: 37,036–28,952,524) sequences. Similarly, the publicly available faecal samples resulted in 15,297,318 (mean: 637,388; range: 151,346–2,703,744) sequences while the neonatal intensive care unit (NICU) and skin samples resulted in 2,589,226 (mean: 92,472; range: 30,690–331,560) and 4,173,296 (mean: 52,166; range: 7,392–185,588) sequences, respectively. A negative control sample analysed in parallel was found to containing only seven 16S rDNA sequences. Open reference OUT clustering (99%) of the complete dataset of 136,935,512 quality filtered sequences, followed by exclusion of those OTUs with fewer than 2 sequences, chimeras and sequences that fail to align with PyNAST [34]; removal of chloroplast, mitochondria and unassigned reads resulted in 1785 OTUs with a total count of 2,382,045 reads.

## Comparison of the microbial composition on textiles and hard surfaces with publicly available data

MicrobiomeAnalyst was used to compare the microbial composition of our sample groups (textile and hard surfaces) with the microbial composition identified in the three previously published 16S rDNA data sets, i.e., skin, faeces, and neonatal intensive care unit. The OTU-data at genus level (Fig 1A) indicated that the composition of the microflora on textiles in the examined cardiac and neonatal care wards were more closely related to skin flora rather than to faecal flora. Surface samples from the published NICU-study did not closely resemble our hard surface samples [26].

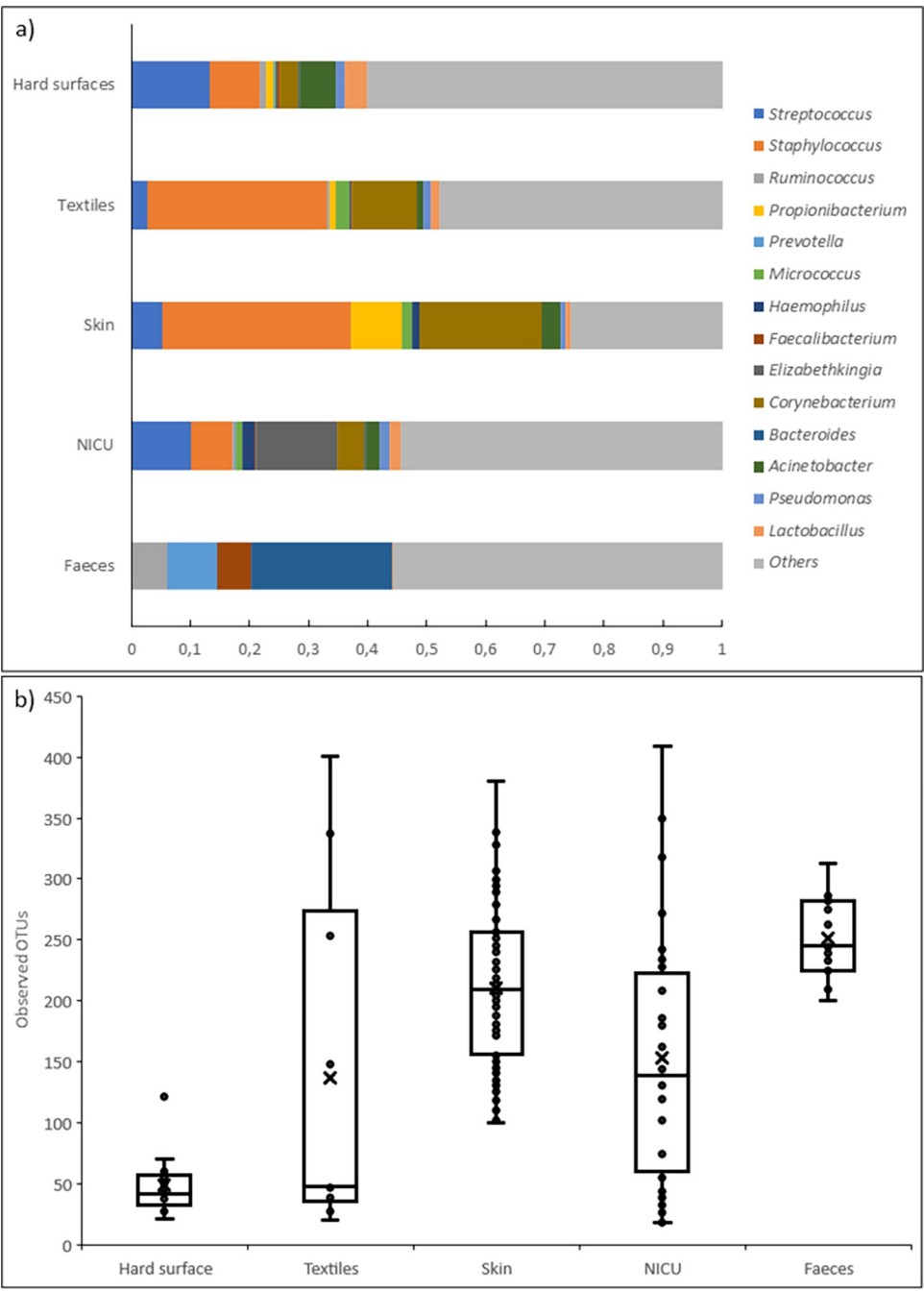

**Fig 1.** a) Top fifteen genera identified by 16S rDNA sequencing analysis in hard surfaces and textiles collected during the study, as well as sequence data from skin, faecal and neonatal intensive care units (NICU) from similar studies [25, 26]. b) The alpha-diversity of the five different sample group-associated bacterial microbiomes presented as community richness (Chao 1).

Several of the identified genera were found to contain both pathogenic and opportunistic bacterial species, e.g., *Staphylococcus*, *Streptococcus*, *Corynebacterium*, *Acinetobacter* and *Pseudomonas*. As shown in Table 2, the relative abundance of *Staphylococcus* (30.4%) and *Corynebacterium* (10.9%) was significantly higher on textiles (p = 0.004 and 0.023, respectively),

**Table 2. Relative abundance (%) of the selected genera found by 16S rDNA sequencing analysis.**

| Genera | Textiles | Hard surfaces | p value |
|---|---|---|---|
| | Mean ± SEM | Mean ± SEM | |
| *Staphylococcus* | 30.4 ± 8.6 | 8.4 ± 1.5 | 0.004 |
| *Streptococcus* | 2.8 ± 0.5 | 13.3 ± 4.7 | 0.095 |
| *Pseudomonas* | 1.2 ± 0.7 | 1.6 ± 0.9 | 0.760 |
| *Corynebacterium* | 10.9 ± 3.8 | 3.2 ± 0.9 | 0.023 |
| *Acinetobacter* | 1.1 ± 0.4 | 6.0 ± 3.7 | 0.306 |

p value vas calculated using unpaired two-tailed t-test

SEM Standard error of the mean

compared to hard surfaces (8.4% and 3.2%, respectively), and similar to the composition of the skin flora that had *Staphylococcus* (32%) and *Corynebacterium* (20.6%) as the most abundant genera. The relative abundance of *Streptococcus* on hard surfaces was found to be 13.3% compared to 2.8% on textiles, although there was no statistically significant difference (p = 0.095). *Acinetobacter* was found at low levels on textiles (1.3%) compared to hard surfaces (0.2%), with no significant difference, whereas *Pseudomonas* was equally present in textiles (1.2%) and hard surfaces (1.6%). *Elizabethkingia* (13.6%) and *Bacteroides* (23.9%) were the most abundant genera for NICU and faecal samples, respectively (Fig 1A).

The alpha-diversity of the five different sample group-associated bacterial microbiomes, presented as community richness (Chao 1), is presented in in Fig 1B. The alpha-diversity estimate suggested that the bacterial community richness within each of the six sample groups differed substantially. The least complex microflora was observed in the hard surface group and the most complex in the faeces group. The Shannon diversity index was 0.518 and the Evenness index was 0.747, showing a higher microbial diversity in textiles compared to hard surfaces. Twenty-six different genera were identified for the hard surfaces, where *Streptococcus* (13.3%) and *Staphylococcus* (8.4%) were the most predominant. The textiles samples contained 96 different genera with *Staphylococcus* (30.4%), *Corynebacterium* (10.9%), *Streptococcus* (2.8%), *Micrococcus* (2.4%), Lactobacillus (1.5%), Pseudomonas (1.2%), *Acinetobacter* and *Propionibacterium* (1.1%, respectively) being the most common.

## Discussion

The identification of bacteria on high-touch sites, including textile and hard surfaces, using microbiological culture methods and 16S rDNA sequencing in two healthcare wards in Sweden gives an overview of the circulating bacteria in the patient environment, provides knowledge regarding reservoirs of pathogens in health care settings, and measures the efficiency of the cleaning routines used in the wards, all of which are factors that can be linked to the spread of infections.

In this study, a lower percentage of the textile samples met the criteria for cleanliness of $\leq 5$ CFU/cm$^2$ for aerobic bacteria and $\leq 1$ CFU/cm$^2$ for *S. aureus*, as compared to hard surfaces, implying that textiles are more probable source of infection compared to hard surfaces. The higher number of bacteria present in the textiles compared to hard surfaces is likely to be correlated with the cleaning routines used in the wards, considering that the changing of drapes occurred every 90 days compared to the daily cleaning of hard surfaces which was made on a daily basis. It is generally assumed that the risk of acquiring infectious bacteria from textiles is low, resulting in less emphasis on the decontamination of textiles compared to non-porous surfaces [35–37]. In addition, there is a lack of standards for monitoring decontamination

efficiency of textiles, with hospitals using diverse methods to decontaminate textiles, as well as widely varying time frames in cleaning routines. Several studies have shown that the survival of potential pathogens on textiles during both domestic and industrial laundering relates to the decontamination process [7, 35, 36].

The 16S rDNA sequencing analysis revealed a high diversity of bacteria present on textiles with *Staphylococcus* (*S. epidermidis*, *S. aureus*, *Staphylococcus* spp.) and *Corynebacterium* as the most common. In contrast, a low diversity of bacteria was found on hard surfaces with *Streptococcus* as the predominant bacteria. This suggests that textiles are a source of contamination and thus a source of infection at hospitals. Teufel *et al.*, showed that the type of textile is linked to the cleanliness, due to the antimicrobial or inherent properties of the textile type to reduce bacterial growth, resulting in a greater variety of taxa on synthetic textiles compared to cellulose-based textiles [38]. Other studies showed that bacterial colonization of textiles depends partially on the hydrophobic and hygroscopic properties of the textile material. Due to the higher hydrophobicity, more bacteria adhered to polyester compared to cotton, resulting in a faster establishment of surface-associated biofilms [39]. For spreading of infections in a hospital environment, the transfer of microorganisms from textiles plays an important role, and key factors including surface properties, friction and moisture of the fabric are important [40]. The transfer of *E. coli*, *S. aureus* and *Bacillus thurengiensis* from cotton or polyester to fingertips was reported to be less efficient than from non-porous surfaces [13, 41]. It is possible that microorganisms become embedded in the matrix of porous surfaces, leading to lower transfer efficiency than from non-porous surfaces, which may reduce their capacity to act as fomites. However, there is also a probability of air-borne transmission in which, for example, bed making may release microorganisms and allow them to settle in the environment [42].

Neither the analysis of culturable bacteria, nor the sequencing analysis of bacterial DNA on the surfaces, revealed significant numbers of Enterobacteriacae, which is an indicator for faecal contamination [43], or *C. difficil*e, which is the most common cause of nosocomial infections and is highly resistant to harsh conditions [44, 45]. Biofilm-forming bacteria are in general more resistant to disinfectants compared to free living bacteria, allowing them to remain on surfaces for longer periods of time. In our study, only a low percentage of biofilm forming bacteria such as *Pseudomonas aeruginosa* and *Acinetobacter baumannii* was found, which could be explained by the fact that the microbiological sampling only included high-touch surfaces, which might not be the surfaces where these bacterial species normally remain. This also explains the dominance of skin microflora species such as *Staphylococcus*, *Micrococcus* and *Streptococcus*.

An advantage of our study is that the microbiological load of the different surfaces was reported together with the identification of the microorganism by sequencing analysis, which gave the possibility not only to identify the microorganisms present on the surfaces, but also to quantify the number of culturable bacteria per sampled area. A combination of both parameters gives the possibility to measure the rate of HCAIs and fomites. Since both textiles and hard surfaces contained multiple species of bacteria commonly associated with the human microflora and no clear evidence of infectious bacteria was found, it is not possible to make clear conclusions for the source of HCAIs. Our findings correlate to previous studies where around one-third of the microorganisms isolated from textiles were from the skin flora of the participants rather than healthcare-associated pathogens [46]. Nevertheless, the fact that the contacts with textiles were more frequent than contacts with hard surfaces, suggests that textiles containing infectious bacteria could be a source of HCAIs.

The role of the inanimate environment is controversial and there is not a strong proof of the environment-to-patient transmission [47]. To demonstrate this correlation, several parameters need to be considered, including the degree of contamination of the nosocomial

environment by specific pathogens, whether the environment was contaminated before or after the patient colonization and if hand hygiene and fomite cleaning was properly performed [45]. To our knowledge, the microbiological analysis performed in this study is unique in that it included several high-touch fabrics that are not normally examined for contamination on a routine basis. Our study shows the degree of contamination and the quality of cleaning of the surfaces, but it does not present evidence to support that this contamination is linked to HCAI transmission routes.

Two limitations were identified for the study. The first is that it did not include an antibiotic resistance analysis for the isolates, which can lead to an incomplete understanding of resistance reservoirs, ineffective infection control strategies, missed opportunities for source identification and transmission routes [48, 49]. The second limitation of the study was that the 16S rDNA sequencing analysis was performed to the genus level and not to the species level due to the low amount of reads in the samples. Since only a low number of species could be identified, a more comprehensive statistical analysis was performed to the genus level. To assess the potential risks associated with specific pathogenic bacteria further studies need to be performed, identifying bacteria on a species level.

## Conclusions

Textiles in the hospital environment constitute an important source of bacterial contamination, with a microbiological load and composition which is significantly different to that found on hard surfaces. Further studies are needed to understand how these surfaces are related to HCAI, the survival time of bacteria in different textiles, and what preventive measures should be taken to increase the proportion of textiles that meet the cleanliness standards.

## Supporting information

**S1 Table. Description of the 26 samples analysed by 16S rDNA sequencing.**
(PDF)

## Acknowledgments

The authors would like to thank the laboratory staff, specifically Lisbeth Märs at RISE for technical assistance, and the hospital staff for good collaboration.

## Author Contributions

**Conceptualization:** Jenny Logenius, Ulrika Husmark, Birgitta Bergström.

**Data curation:** Erik Nygren, Lucia Gonzales Strömberg.

**Formal analysis:** Erik Nygren, Lucia Gonzales Strömberg.

**Funding acquisition:** Jenny Logenius, Ulrika Husmark, Birgitta Bergström.

**Investigation:** Erik Nygren.

**Project administration:** Birgitta Bergström.

**Visualization:** Charlotta Löfström.

**Writing – original draft:** Lucia Gonzales Strömberg.

**Writing – review & editing:** Charlotta Löfström.

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
