## [Decision Letter · Decision Letter 0]

20 Apr 2023

PONE-D-23-01699Potential sources of contamination on textiles and hard surfaces identified as high-touch sites near the patient environmentPLOS ONE

Dear Dr. Gonzales Strömberg,

Thank you for submitting your manuscript to PLOS ONE. After careful consideration, we feel that it has merit but does not fully meet PLOS ONE’s publication criteria as it currently stands. Therefore, we invite you to submit a revised version of the manuscript that addresses the points raised during the review process.

We look forward to receiving your revised manuscript.

Kind regards,

Awatif Abid Al-Judaibi, PhD

Academic Editor

PLOS ONE

Reviewers' comments:

Reviewer's Responses to Questions

**Comments to the Author**

1. Is the manuscript technically sound, and do the data support the conclusions?

Reviewer #1: Yes

Reviewer #2: Partly

Reviewer #3: Yes

2. Has the statistical analysis been performed appropriately and rigorously? 

Reviewer #1: Yes

Reviewer #2: Yes

Reviewer #3: Yes

3. Have the authors made all data underlying the findings in their manuscript fully available?

Reviewer #1: No

Reviewer #2: No

Reviewer #3: Yes

4. Is the manuscript presented in an intelligible fashion and written in standard English?

Reviewer #1: Yes

Reviewer #2: No

Reviewer #3: Yes

5. Review Comments to the Author

Reviewer #1: Although the authors said they have provided the data, they only provided the numbers of the 26 analysed samples. This is still missing data. is it possible to access the actual data analysed as opposed to the numbers in the sections of data availability.

Reviewer #2: PLOS one review

Potential sources of contamination on textiles and hard surfaces identified as high-

touch sites near the patient environment

Introduction

Line 48:

Even though, not Even tough

Methods

Comment: The method section can be enhanced by breaking some of the subsections further. For instance, the first subsection ‘Study environment and observational study (now study design)’ can be divided into 2 separate sections.

Observational study is too broad, and it comprises several study designs such as case reports, case series, cross-sectional study, etc (https://www.ncbi.nlm.nih.gov/pmc/articles/PMC6003013/). I suggest that the authors be specific with their study design, and replace all observational study with the specific design.

Line 87:

‘Study environment and study design’ is more appropriate

Line 89:

Replace ‘during’ with ‘from’

Line 101 – 105:

I suggest this paragraph stand alone as a subsection titled ‘Ethical Approval’. Also, I suggest you provide approval number from the committee.

Line 107 – 141:

As mentioned previously, break this subsection further. So, the authors can talk about sample collection alone, then sample preparation, etc…

Not antibiotic susceptibility tests performed???

Line 123:

Rinsing with what? Just water, or soapy water??

Line 168:

I suggest significant value be p< 0.05, and not p=0.05.

Results

Line 172 – 176:

This paragraph is not necessary under results. The authors should only talk about the results obtained after they carried out their experiment, and not to give explanations or preambles.

So, from the results, there was no C. difficile isolated and identified during the culture phase. This should be stated clearly.

Table 1:

I suggest the authors define the values inside the bracket/parenthesis. Since n=176 for both columns, you can include the ‘n=176’ at the end of the table 1 title. Then, you can define the column as n(%), meaning you have the values on the outside and their respective percentages inside the parenthesis. Don’t mixed the decimal points up (be consistent with the ‘,’ or ‘.’).

Table 2:

Wondering why the authors didn’t identify further to the species level. We know not all Staphylococcus, Strep, Pseudo etc are normal flora. So, speciation would have given us more insight into which category of organisms we are dealing with: whether pathogenic types or not.

Line 206:

In addition to, not ‘in additional to’

Line 335:

Lack of evidence. Insert ‘of’.

The authors should let us know if the study had any limitation(s). This can be part of the discussion.

General comments:

1. I know multidrug resistance is a big problem in hospital settings. Why didn’t the authors perform antibiotic susceptibility tests on the isolated microorganisms?

2. I suggest authors get a native English speaker to proof-read the entire work and make necessary corrections to all erroneous spellings, punctuations and grammatical expressions.

Reviewer #3: THE STUDY WAS WELL-THOUGHT THROUGH AND CAREFULLY EXECUTED. I HOWEVER, HAVE THESE FEW ISSUES/CONCERNS THAT THE AUTHORS MAY CONSIDER ADDRESSING

GENERAL CONCERNS

1. What type of study design was employed here?

2. How was sterility of the sponges used for the sampling assured; was there any control (negative control) for the culturable samples. If this was not done, it should be stated as limitation of the work.

3. What informed the sample size of 176 (113 from hard non-porous surfaces and 63 from textile surfaces)

4. For reproducibility purposes, what was the time period between cleaning of the surfaces and sampling

5. A brief summary of how the various bacterial isolation or identification was performed/made would be desirable

SPECIFICS CONCERNS

1. “regarding the number of bacteria and the bacterial community composition” of 1st statement under Sample Collection, sample preparation and microbial analysis is not necessary

2. Second statement under same section stated just above needs clarification and revision – are you describing sites or surfaces where samples were taken?

3. Second paragraph of same section “Triplicate samples were collected from the samples that were…” Is surface interchanged with sample?

4. Third paragraph of same section, provide unit for “15”. Also, same paragraph, be clear whether the supernatant was discarded, and pellets/sediment re-suspended in peptone water.

6. PLOS authors have the option to publish the peer review history of their article (what does this mean?). If published, this will include your full peer review and any attached files.

Reviewer #1: No

Reviewer #2: No

Reviewer #3: **Yes: **FRANCIS OPOKU AGYAPONG

---

## [Author Response · Author response to Decision Letter 0]

31 May 2023

Reviewer 1.

Introduction. Line 48: Even though, not Even tough

The correction has been implemented in the text.

Methods. Comment: The method section can be enhanced by breaking some of the subsections further. For instance, the first subsection ‘Study environment and observational study (now study design)’ can be divided into 2 separate sections. Observational study is too broad, and it comprises several study designs such as case reports, case series, cross-sectional study, etc (https://www.ncbi.nlm.nih.gov/pmc/articles/PMC6003013/). I suggest that the authors be specific with their study design and replace all observational study with the specific design.

Following the suggestion and based on the literature, we have modified the observational study to a cross-sectional study in the text (lines 88-101). 

Line 87: ‘Study environment and study design’ is more appropriate.

The suggestion has been implemented in the text.

Line 89: Replace ‘during’ with ‘from’.

The correction has been implemented in the text.

Line 101 – 105: I suggest this paragraph stand alone as a subsection titled ‘Ethical Approval’. Also, I suggest you provide approval number from the committee.

The suggestion has been implemented in the text. We do not have approval number from the committee, just a notification from the committee. It is important to mention that surface and fabric samples collected in this study did not qualify as biological samples according to the Swedish ethical law.

Line 107 – 141: 

As mentioned previously, break this subsection further. So, the authors can talk about sample collection alone, then sample preparation, etc…

The suggestion has been implemented. Now there is separate sections for the study design (line 87), ethical approval (line 103), sample collection (line 110), sample preparation (line 132) and microbial culture analysis (line 141). 

Not antibiotic susceptibility tests performed???

No antibiotic susceptibility test was performed in the frame of the study, and we acknowledge that this is a limitation of the study. A paragraph has been included in the discussion section to lift this point (lines 342-345). Unfortunately, since we only performed 16S rDNA sequencing it was not possible to look at a later stage for antibiotic resistance markers in the sequences. 

Line 123: Rinsing with what? Just water, or soapy water??

A clarification was added in the text; the samples were rinsed with peptone water. 

Line 168: I suggest significant value be p< 0.05, and not p=0.05. 

The suggestion has been implemented in the text. 

Results Line 172 – 176: This paragraph is not necessary under results. The authors should only talk about the results obtained after they carried out their experiment, and not to give explanations or preambles.

The paragraph has been removed from the results section and the information has been included in the methods (lines 99-101).

So, from the results, there was no C. difficile isolated and identified during the culture phase. This should be stated clearly.

A clarification of this statement was included in the text (line 201). 

Table 1: I suggest the authors define the values inside the bracket/parenthesis. Since n=176 for both columns, you can include the ‘n=176’ at the end of the table 1 title. Then, you can define the column as n(%), meaning you have the values on the outside and their respective percentages inside the parenthesis. Don’t mixed the decimal points up (be consistent with the ‘,’ or ‘.’). 

The suggestions have been implemented in the table. The table heading has been rewritten to improve clarity.

Table 2: Wondering why the authors didn’t identify further to the species level. We know not all Staphylococcus, Strep, Pseudo etc are normal flora. So, speciation would have given us more insight into which category of organisms we are dealing with whether pathogenic types or not. 

Unfortunately, the resolution of the sequencing was not optimal for identification to the species levels but enough to discriminate to the genus level. For example, for Staphylococcus, species identification was reached for S. epidermidis, but a significant amount of the reads only identified Staphylococcus spp. In this context no statistical analysis was possible and therefore the analysis was performed to the genus level. Same pattern was observed for the other genera. We acknowledge that is a drawback of the study and has been included in the discussion section (lines 345-350). 

Line 206: In addition to, not ‘in additional to’

The suggestion has been implemented in the text. 

Line 335: Lack of evidence. Insert ‘of’.

The suggestion has been implemented in the text. 

The authors should let us know if the study had any limitation(s). This can be part of the discussion.

A paragraph about the limitations of the study including the lack of antibiotic resistance analysis as well as the sequencing analysis to the genus level and not to the species level was included (345-350).

 

Reviewer #2: 

I know multidrug resistance is a big problem in hospital settings. Why didn’t the authors perform antibiotic susceptibility tests on the isolated microorganisms? 

Following the comment from Reviewer #1 about the same issue, this has been addressed as a limitation of our study in the discussion section (lines 342-345). To clarify, detection of antibiotic resistance was not in the scope of the study and therefore no following testing could be performed. In addition, since the sequencing analysis was done using 16S rDNA sequencing, it was not possible to perform the analysis in the sequence level. 

2. I suggest authors get a native English speaker to proof-read the entire work and make necessary corrections to all erroneous spellings, punctuations and grammatical expressions.

The revised manuscript has been proof-red by a native English speaker. 

Reviewer #3: 

The study was well-thought through and carefully executed. I, however, have these few issues/concerns that the authors may consider addressing

1. What type of study design was employed here?

Following the suggestion of Reviewer 1, a clarification of the type of the observational study implemented in this study was included in the study design section in the methods. A cross-sectional study was performed in this study. 

2. How was sterility of the sponges used for the sampling assured; was there any control (negative control) for the culturable samples. If this was not done, it should be stated as limitation of the work.

The sponges are sterile from the factory. A negative control was included in this study and a clarification of this was included in the method section, lines 124-125. 

3. What informed the sample size of 176 (113 from hard non-porous surfaces and 63 from textile surfaces)

The sample size was identified based on the number of high hand touch surfaces identified during the time of the study. A clarification was included in lines 195-197. 

4. For reproducibility purposes, what was the time period between cleaning of the surfaces and sampling.

The time between cleaning of the surfaces and sampling was 20 minutes. This information has been included in the method section (lines 119-120).

5. A brief summary of how the various bacterial isolation or identification was performed/made would be desirable.

The samples were plated on specific agar media for each of the bacteria of interest. TSA agar plates were used to determine aerobic bacteria, Violet Red Bile Glucose (VRBG) plates for Enterobacteriacea, Baird Parker Agar for S. aureus and Clostridium difficile Selective Agar for C. difficile. Presence of growth in each of the plates was reported positive for the given bacteria and quantification was performed for each plate when growth was registered. 

SPECIFICS CONCERNS

1. “regarding the number of bacteria and the bacterial community composition” of 1st statement under Sample Collection, sample preparation and microbial analysis is not necessary.

We disagree with the comment. We consider this paragraph important to understand the study design and the considerations considered in the study. 

2. Second statement under same section stated just above needs clarification and revision – are you describing sites or surfaces where samples were taken?

Samples were taken from rinsing textiles and swabbing surfaces. Both textiles and surfaces were identified in the study as high-touch objects. The text has been clarified to better describe the sampling procedure (lines 133-139). 

3. Second paragraph of same section “Triplicate samples were collected from the samples that were…” Is surface interchanged with sample?

The sentence has been rewritten to improve clarity (line 119). Also refer to the previous answer that elaborates on the sampling procedure. 

4. Third paragraph of same section, provide unit for “15”. Also, same paragraph, be clear whether the supernatant was discarded, and pellets/sediment re-suspended in peptone water.

A clarification was included, the supernatant was discarded, and the pellet was resuspended in 300 µl of peptone water (line 135-137).

---

## [Decision Letter · Decision Letter 1]

14 Jun 2023

Potential sources of contamination on textiles and hard surfaces identified as high-touch sites near the patient environment

PONE-D-23-01699R1

Dear Dr. Lucia Gonzales Strömberg,

We’re pleased to inform you that your manuscript has been judged scientifically suitable for publication and will be formally accepted for publication once it meets all outstanding technical requirements.

Kind regards,

Awatif Abid Al-Judaibi, PhD

Academic Editor

PLOS ONE

Dear authors,

Thank you for your plosive responses to authors, Although I have given an "accept" for publishing, I recommend following the reviewer's comment:

The authors have addressed most of concerns raised earlier. My only reservation is the phrase "Cross-sectional Study" immediately following "results" at the results section. The entire study design including the laboratory work is cross-sectional in nature. Therefore, singling out the results on "Frequency of direct surface contacts/touch" as 'cross-sectional study' may be misleading.

Reviewers' comments:

Reviewer's Responses to Questions

**Comments to the Author**

1. If the authors have adequately addressed your comments raised in a previous round of review and you feel that this manuscript is now acceptable for publication, you may indicate that here to bypass the “Comments to the Author” section, enter your conflict of interest statement in the “Confidential to Editor” section, and submit your "Accept" recommendation.

Reviewer #1: All comments have been addressed

Reviewer #2: All comments have been addressed

Reviewer #3: All comments have been addressed

2. Is the manuscript technically sound, and do the data support the conclusions?

Reviewer #1: Yes

Reviewer #2: Yes

Reviewer #3: Yes

3. Has the statistical analysis been performed appropriately and rigorously? 

Reviewer #1: Yes

Reviewer #2: Yes

Reviewer #3: Yes

4. Have the authors made all data underlying the findings in their manuscript fully available?

Reviewer #1: Yes

Reviewer #2: Yes

Reviewer #3: Yes

5. Is the manuscript presented in an intelligible fashion and written in standard English?

Reviewer #1: Yes

Reviewer #2: Yes

Reviewer #3: Yes

6. Review Comments to the Author

Reviewer #1: The author has ably addressed the concerns and I forward to you for publication. all the concerns that I raised have been satisfactorily addressed.

Reviewer #2: At this point, I do not have any further comments.

All comments have been answered by the authors.

Comments authors were unable to address have been stated clearly as study limitation(s).

Reviewer #3: The authors have addressed most of concerns raised earlier. My only reservation is the phrase "Cross-sectional Study" immediately following "results" at the results section. The entire study design including the laboratory work is cross-sectional in nature. Therefore, singling out the results on "Frequency of direct surface contacts/touch" as 'cross-sectional study' may be misleading.

7. PLOS authors have the option to publish the peer review history of their article (what does this mean?). If published, this will include your full peer review and any attached files.

Reviewer #1: No

Reviewer #2: No

Reviewer #3: No

---

## [Editor Report · Acceptance letter]

28 Jun 2023

PONE-D-23-01699R1 

Potential sources of contamination on textiles and hard surfaces identified as high-touch sites near the patient environment. 

Dear Dr. Gonzales Strömberg:

I'm pleased to inform you that your manuscript has been deemed suitable for publication in PLOS ONE. Congratulations! Your manuscript is now with our production department. 

Kind regards, 

on behalf of

Professor Awatif Abid Al-Judaibi 

Academic Editor

PLOS ONE